# Changes in microRNA Expression in the Cochlear Nucleus and Inferior Colliculus after Acute Noise-Induced Hearing Loss

**DOI:** 10.3390/ijms21228792

**Published:** 2020-11-20

**Authors:** Sohyeon Park, Seung Hee Han, Byeong-Gon Kim, Myung-Whan Suh, Jun Ho Lee, Seung Ha Oh, Moo Kyun Park

**Affiliations:** 1Interdisciplinary Program in Neuroscience, Seoul National University College of Natural Sciences, Seoul 08826, Korea; 5liviapark@snu.ac.kr; 2Department of Otorhinolaryngology-Head and Neck Surgery, Seoul National University College of Medicine, Seoul 03080, Korea; hee92h@nate.com (S.H.H.); byeonggone@naver.com (B.-G.K.); drmung@naver.com (M.-W.S.); junlee@snu.ac.kr (J.H.L.); shaoh@snu.ac.kr (S.H.O.); 3Sensory Organ Research Institute, Seoul National University Medical Research Center, Seoul 03080, Korea; 4Seoul Wide River Institute of Immunology, Seoul National University, Gangwon 25159, Korea

**Keywords:** noise-induced hearing loss, microRNAs, cochlear nucleus, inferior colliculus, neuroplasticity

## Abstract

Noise-induced hearing loss (NIHL) can lead to secondary changes that induce neural plasticity in the central auditory pathway. These changes include decreases in the number of synapses, the degeneration of auditory nerve fibers, and reorganization of the cochlear nucleus (CN) and inferior colliculus (IC) in the brain. This study investigated the role of microRNAs (miRNAs) in the neural plasticity of the central auditory pathway after acute NIHL. Male Sprague–Dawley rats were exposed to white band noise at 115 dB for 2 h, and the auditory brainstem response (ABR) and morphology of the organ of Corti were evaluated on days 1 and 3. Following noise exposure, the ABR threshold shift was significantly smaller in the day 3 group, while wave II amplitudes were significantly larger in the day 3 group compared to the day 1 group. The organ of Corti on the basal turn showed evidence of damage and the number of surviving outer hair cells was significantly lower in the basal and middle turn areas of the hearing loss groups relative to controls. Five and three candidate miRNAs for each CN and IC were selected based on microarray analysis and quantitative reverse transcription PCR (RT-qPCR). The data confirmed that even short-term acoustic stimulation can lead to changes in neuroplasticity. Further studies are needed to validate the role of these candidate miRNAs. Such miRNAs may be used in the early diagnosis and treatment of neural plasticity of the central auditory pathway after acute NIHL.

## 1. Introduction

According to the World Health Organization, 466 million people—more than 5% of the world’s population—are affected by hearing loss [1]. Loss of hearing makes communication difficult and can lead to psychosocial problems, such as depression and feelings of loneliness [2]. Therefore, losing the ability to hear can affect people’s lives in many ways.

NIHL is a type of acquired hearing loss that is due to a sudden excessively loud sound or continuous moderately loud sounds. Depending on the duration and intensity of the sounds, hearing loss can be temporary or permanent [3]. Threshold sensitivity loss after exposure to a loud noise can be recovered to baseline levels after a few hours to weeks and that is called temporary threshold shift (TTS). Permanent threshold shift (PTS) occurs when the noise is too loud to recover to its baseline level or the noise takes place repeatedly or continuously, impeding recovery.

Even though hearing thresholds may return to normal, the changes in the synapse may affect auditory processing and hearing upon noise [4]. Exposure to noise not only targets the hair cells (HCs) in the cochlea, but also the neuronal cells in the auditory pathway. Noise exposure can also alter the synaptic transmission and ion channel function, distort sensory maps, and cause abnormal neuronal firing patterns. These cellular and physiological changes can cause another disorder, such as loudness recruitment or tinnitus [5].

NIHL usually progresses slowly over a period of years, making it difficult to recognize. Initial symptoms include difficulty in differentiating conversations against background noise. Later, listening becomes difficult, even in ordinary circumstances. Since NIHL progresses gradually and prevention is so crucial, it is important to diagnose NIHL at an early stage [6].

MicroRNAs (MiRNAs) are small non-coding single-stranded endogenous RNAs that regulate posttranscriptional gene expression. Mature miRNAs are 18–22 nucleotides in length and can repress and degrade target mRNAs by binding to their 3′ or 5′ untranslated regions. MiRNAs are involved in a number of physiological processes and play key roles in neural development and plasticity. MiRNAs are abundant in the brain, due to its complexity [7,8], and are also found in body fluids such as the cerebrospinal fluid, whole blood, plasma, and serum [9]. Consequently, miRNAs can be detected and analyzed non-surgically and may provide the basis for a new generation of therapeutic agents [10]. In fact, there are some miRNA-based therapeutics for cancer and other diseases and several miRNAs are currently being studied in Phase I/II clinical trials, with promising outcomes [11].

Electrical signals from the HCs in the inner ear flow to primary sensory neurons, which are called spiral ganglion neurons. These cochlear nerves combine with vestibular nerves to become branch VIII of the cranial nerve [12], which transfers the converted acoustic information to the cochlear nucleus (CN), developing the neural pathway [13]. The superior olivary complex (SOC) in the ventral auditory brainstem receives auditory information from the CN and sends the signal to the inferior colliculus (IC). The IC plays a crucial role in the auditory pathway and is the first location where parallel auditory signals from both sides of the cochlea are integrated. Most of the auditory information is processed in the IC and sent to the auditory cortex through the medial geniculate body [14].

Noise exposure is one of the most common causes of hearing loss. Exposure to loud noise leads to secondary changes, such as decreases in the number of synapses, the degeneration of auditory nerve fibers, and reorganization of the CN and IC, which may induce neural plasticity in the central auditory pathway. However, little is known about the role of miRNAs in the central auditory pathway. Therefore, this study investigated the role of miRNAs in the neural plasticity of the central auditory pathway after NIHL.

## 2. Results

### 2.1. Hearing Changes after Noise Exposure

Tone-burst acoustic stimuli were measured at three different frequencies of 4, 8, and 16 kHz. Our noise-control model (*n* = 4) showed that the auditory brainstem response (ABR) threshold started to decrease at day 1 after the noise exposure. By day 3 after the exposure to noise at 4 and 8 kHz, the ABR thresholds had returned to normal (Figure 1a). The mean ABR thresholds observed for the day 1 control group (4 kHz, 20.6 ± 2.2 dB; 8 kHz, 21.3 ± 2.7 dB; 16 kHz, 25.4 ± 3.6 dB), day 1 group (4 kHz, 20.0 ± 0.0 dB; 8 kHz, 20.8 ± 1.9 dB; 16 kHz, 27.1 ± 3.6 dB), day 3 control group (4 kHz, 20.8 ± 2.4 dB; 8 kHz, 21.9 ± 3.2 dB; 16 kHz, 25.6 ± 4.3 dB), and day 3 group (4 kHz, 24.2 ± 4.8 dB; 8 kHz, 25.8 ± 4.6 dB; 16 kHz, 26.9 ± 3.6 dB) confirmed that hearing levels at all three frequencies were normal before the experiment (Figure 1b–d). After 2 h of noise exposure, both the day 1 group (4 kHz, 81.9 ± 11.6 dB; 8 kHz, 87.1 ± 3.3 dB; 16 kHz, 88.3 ± 2.4 dB) and the day 3 group (4 kHz, 78.8 ± 11.8 dB; 8 kHz, 84.6 ± 2.9 dB; 16 kHz, 86.7 ± 3.8 dB) exhibited significant increases in ABR thresholds compared to the day 1 control group (4 kHz, 20.2 ± 1.0 dB; 8 kHz, 20.4 ± 1.4 dB; 16 kHz, 23.5 ± 3.8 dB) and the day 3 control group (4 kHz, 20.4 ± 1.4 dB; 8 kHz, 21.0 ± 2.5 dB; 16 kHz, 24.4 ± 3.7 dB), respectively (Figure 1b–d). At 4 kHz, the mean ABR threshold of the day 1 group (65.6 ± 19.5 dB) was slightly lower than that of the day 1 control group (81.9 ± 11.6 dB). Moreover, the mean ABR threshold of the day 3 group (42.7 ± 17.1 dB) was lower than that of the day 1 group. The differences between these groups were significant (*p* < 0.001; Figure 1b,e). At 8 kHz, the mean ABR thresholds of both the day 1 (73.3 ± 10.7 dB) and day 3 (51.9 ± 13.7 dB) groups were significantly lower than those of the control groups (87.1 ± 3.3 dB; *p* < 0.001; Figure 1c,f). The mean ABR thresholds of the day 1 (79.4 ± 8.5 dB) and day 3 (63.5 ± 12.6 dB) groups were also significantly lower than those of the control groups (88.3 ± 2.4 dB) at 16 kHz (*p* < 0.001; Figure 1d,g). The day 3 group exhibited significantly better hearing than the day 1 group at all three frequencies (*p* < 0.001; Figure 1e–g).

### 2.2. ABR Amplitudes

The amplitudes of waves II and IV were calculated based on ABR waveforms. The wave II amplitudes observed in the day 1 treatment group (4 kHz, 0.61 ± 0.28 µV; 8 kHz, 0.62 ± 0.25 µV; 16 kHz, 0.39 ± 0.23 µV) were significantly smaller than those observed in the day 3 treatment group, at all frequencies (4 kHz, 1.56 ± 0.90 µV; 8 kHz, 1.45 ± 0.84 µV; 16 kHz, 1.08 ± 0.0 µV; *p* < 0.001; Figure 2a). At 4 kHz, the mean wave IV amplitude observed in the day 1 treatment group (0.44 ± 0.39 µV) was significantly smaller than that in the day 3 treatment group (0.93 ± 0.22 µV). However, at the other frequencies, the amplitudes observed in the day 1 treatment group (8 kHz, 0.36 ± 0.22 µV; 16 kHz, 0.26 ± 0.18 µV) did not differ significantly from those observed on day 3 (8 kHz, 0.43 ± 0.40 µV; 16 kHz, 0.26 ± 0.26; Figure 2b).

### 2.3. ABR Latencies

Latencies between waves IV and II were calculated based on ABR waveforms. At 4 kHz, there was no difference in latencies between the day 1 (2.54 ± 0.50 ms) and day 3 treatment groups (2.64 ± 0.77 ms). Similar results were seen at 16 kHz, with no difference in latencies being evident between the day 1 (2.77 ± 0.15 ms) and day 3 treatment groups (2.80 ± 0.41 ms). Conversely, at 8 kHz, the latency observed in the day 3 treatment group (2.38 ± 0.54 ms) was slightly reduced compared to that of the day 1 treatment group (2.68 ± 0.71 ms); however, this difference was not statistically significant (Figure 2c).

### 2.4. Histology of the Organ of Corti

A series of sagittal sections from decalcified cochleae were stained with hematoxylin and eosin (H&E) to investigate the structure of the organ of Corti. These structures were intact in both the day 1 and day 3 control groups. The apical turn and middle turn sections of the organ of Corti from the day 1 and day 3 treatment groups were also normal, with intact HCs and other non-sensory cells, such as Deiter and pillar cells. However, the basal turn sections of the organs of Corti exhibited abnormalities. The degree of damage differed among cochleae, with some exhibiting only a loss of HCs, and others also exhibiting a loss of supporting cells (Figure 3).

### 2.5. Phalloidin Staining of Outer HCs

Following completion of the whole mount surface preparation procedure, phalloidin was used to stain the outer HCs. Two 200-µm-long segments were selected from each turn section of the organ of Corti, and the mean number of surviving outer HCs was determined. All three rows of outer HCs from all control samples were normal, with no evidence of missing HCs. However, significant HC loss was observed in the basal turn sections in both the day 1 (85% ± 7%; *p* = 0.001) and day 3 (93% ± 6%; *p* = 0.019) treatment groups. Some HCs were also lost from the middle turn sections of samples from the day 1 (99% ± 1%; *p* = 0.008) and day 3 (99% ± 1%; *p* < 0.001) treatment groups. In contrast, outer HCs in the apical turn sections of samples from both the day 1 and day 3 treatment groups were largely intact (both, 100% ± 0%). For the basal turn sections, significantly fewer HCs were lost from the day 3 treatment group compared to the day 1 treatment group (*p* = 0.039). No statistically significant difference in HC loss was observed between the day 1 and day 3 groups for the middle or apical turn sections of the organ of Corti (Figure 4).

### 2.6. Selection of Candidate miRNAs

#### 2.6.1. The CN

Microarray analysis of the CN identified 1228 candidate miRNAs. Changes in miRNA expression were assessed via three pair-wise comparisons: Between the day 1 treatment and control groups; between the day 3 treatment and control groups; and between the day 1 and day 3 groups. For each comparison, miRNAs with normalized expression changes ≥1.5-fold (*p* < 0.1) were excluded. Then, the miRNAs from each set were combined and those found in more than one dataset were eliminated. miRNAs that exhibited differences in expression between the day 1 and day 3 groups were also excluded. A hierarchical clustering heat map was created using Multiple Experiment Viewer (http://mev.tm4.org) to visualize the remaining 33 miRNAs (Figure 5a). Then, miRNAs with ≥1.5-fold differences in expression between the day 1 and day 3 control groups were excluded. Of the 21 miRNAs that remained, only those expressed in humans were considered for further analysis, resulting in a final list of 10 candidate miRNAs, including *miR-411-3p*, *miR-183-5p*, *miR-377-3p*, *miR-20b-5p*, *miR-137-5p*, *miR-211-3p*, *miR-483-5p*, *miR-92a-1-5p*, *miR-187-5p*, and *miR-200b-3p* (Table 1).

#### 2.6.2. The IC

Microarray analysis of the IC identified 1200 miRNAs. Changes in miRNA expression were assessed in three pair-wise comparisons: Between the day 1 treatment and control groups; between the day 3 treatment and control groups; and between the day 1 and day 3 groups. For each comparison, miRNAs with ≥1.5-fold normalized expression changes (*p* < 0.1) were excluded. Then, the miRNAs from each set were combined, and those found in more than one dataset were eliminated. miRNAs that exhibited differences in expression between the day 1 and day 3 groups were also excluded. A hierarchical clustering heat map was created using Multiple Experiment Viewer to assess the remaining 27 miRNAs (Figure 5b). Then, miRNAs with expression differences changes >1.5-fold or <0.5-fold between the day 3 and day 1 control groups were excluded. Of the 26 miRNAs that remained, only those expressed in humans were considered for further analysis, resulting in a final list of 13 candidate miRNAs, including *miR-204-5p*, *miR-376b-5p*, *miR-26b-5p*, *miR-136-3p*, *miR-132-5p*, *miR-128-2-5p*, *miR-132-3p*, *miR-377-5p*, *miR-210-3p*, *miR-92a-1-5p*, *miR-425-3p*, *miR-362-5p*, and *miR-150-3p* (Table 2).

### 2.7. Validation of Candidate miRNAs Using qRT-PCR

#### 2.7.1. The CN

Based on the results of the microarray analysis of the CN, 10 candidate miRNAs were selected for further analysis, including *miR-411-3p*, *miR-183-5p*, *miR-377-3p*, *miR-20b-5p*, *miR-137-5p*, *miR-211-3p*, *miR-483-5p*, *miR-92a-1-5p*, *miR-187-5p*, and *miR-200b-3p* (Figure 6a). Microarray data of the day 3 and day 1 treatment groups are compared in Figure 6b, along with the accompanying qRT-PCR results. Based on these data, five miRNAs were selected due to their consistent expression patterns, including *miR-411-3p*, *miR-183-5p*, *miR-377-3p*, *miR-20b-5p*, and *miR-200b-3p*. The expression of *miR-200b-3p* increased after noise exposure, whereas that of the other miRNAs decreased.

#### 2.7.2. The IC

Based on the results of the microarray analysis of the IC, 13 candidate miRNAs were selected for further analysis, including *miR-204-5p*, *miR-376b-5p*, *miR-26b-5p*, *miR-136-3p*, *miR-132-5p*, *miR-128-2-5p*, *miR-132-3p*, *miR-377-5p*, *miR-210-3p*, *miR-92a-1-5p*, *miR-425-3p*, *miR-362-5p*, and *miR-150-3p* (Figure 7a). Microarray data of the day 3 and day 1 treatment groups are compared in Figure 7b, along with the accompanying qRT-PCR results. Three miRNAs were selected due to their consistent expression patterns, including *miR-92a-1-5p*, *miR-136-3p*, and *miR-26b-5p*. The expression of *miR-92a-1-5p* increased after noise exposure, whereas that of the other miRNAs decreased.

### 2.8. Target Pathway Analysis of Candidate miRNAs

#### 2.8.1. The CN

Five candidate miRNAs expressed in the CN were validated using qRT-PCR, including *miR-411-3p*, *miR-183-5p*, *miR-377-3p*, *miR-20b-5p*, and *miR-200b-3p*. DIANA-miRPath software (ver. 3.0; http://www.microrna.gr/miRPathv3) was used to investigate the regulation of biological pathways by miRNAs in the CN. A Kyoto Encyclopedia of Genes and Genomes (KEGG) analysis identified 12 significantly overrepresented pathways. The most relevant pathways for these miRNAs involved mitogen-activated protein kinase (MAPK) signaling, axon guidance, and transforming growth factor-beta (TGF-β) signaling (Figure 8a).

#### 2.8.2. The IC

Three candidate miRNAs expressed in the IC were validated using qRT-PCR, including *miR-92a-1-5p, miR-136-3p*, and *miR-26b-5p*. DIANA-miRPath software (ver. 3.0; DIANA TOOLS, http://www.microrna.gr/miRPathv3) was used to investigate the regulation of biological pathways by miRNAs in the IC. A KEGG analysis identified 14 significantly overrepresented pathways. The most relevant pathway for these three miRNAs was the MAPK signaling pathway (Figure 8b).

## 3. Discussion

Our results demonstrated that even short-term acoustic stimulation can cause changes in miRNA expression in the CN and IC, and that these changes may also induce plasticity in the central auditory pathway. Microarray analysis and qRT-PCR suggested that miRNAs play a key role in neural plasticity after noise exposure in both the CN (i.e., *miR-411-3p*, *miR-183-5p*, *miR-377-3p*, *miR-20b-5p*, and *miR-200b-3p*) and IC (i.e., *miR-92a-1-5p*, *miR-136-3p*, and *miR-26b-5p*). Further research is necessary to understand the specific roles of these candidate miRNAs, with preliminary evidence suggesting that they may be involved in regulating the MAPK signaling pathway, axon guidance, and the TGF-β signaling pathway. Numerous studies have investigated NIHL without identifying a reliable tool for early diagnosis or a treatment that results in complete recovery. Currently, hearing aids and cochlear implants are used to treat patients with severe hearing loss, but more effective methods for diagnosing and treating NIHL are required. Since miRNAs are stable over long periods of time and can be detected in the blood, as well as in brain tissue, these sequences may represent a viable diagnostic target for blood tests, enabling earlier diagnosis of NIHL and potentially protecting people against tinnitus. Moreover, gene therapy involving the transfer of miRNAs to target cells using viral vectors or siRNAs could be used to treat NIHL in the future.

Consecutive ABR tests were performed in this study, with evidence of ABR threshold recovery beginning as early as day 1 after noise exposure. The shift in the ABR threshold decreased significantly on days 2 and 3 after noise exposure, and was insignificant after day 3. Based on these observations, we hypothesized that significant changes may occur within this time frame. We therefore chose day 1 and 3 after noise exposure as our two major time points.

We focused on the CN and IC, because the latter is the origin of the central auditory pathway and the IC forms its core. The central auditory pathway receives the bilateral auditory signal, with waves II and IV corresponding to the CN and IC, respectively [15]. Therefore, we evaluated the latency between waves IV and II, and the height of these waves relative to their resting points. The latency between waves IV and II was slightly reduced at 8 kHz in the day 3 treatment group; however, the difference was not statistically significant. Changes in the latency of wave I might be expected because the auditory nerve is the primary region affected by HC loss [16]. An increased latency of wave I may indicate dysfunction in action potential propagation along the auditory nerve [17]. However, it is difficult to measure wave I using the ABR [18]. Moreover, short-term noise exposure may affect auditory nerve fibers, but not the CN or IC.

We observed a statistically significant difference in the amplitude of wave II between the day 1 and day 3 treatment groups. The amplitude of wave II was greater in the day 3 treatment group at all frequencies. In general, the ABR amplitude of wave I is reduced by overexpressed sound stimuli [19,20], while the amplitudes of later ABR waves (i.e., II–V) should increase due to compensatory hyperactivity of the central auditory pathway. A reduced sensory input from both ears triggers an increase in excitatory activity and/or a decrease in inhibitory activity [21]. Regarding interpretation of the ABR wave, latency represents the speed of transmission and amplitude represents the number of neurons that fire together [22]. One possible explanation for our observations is that the number of neurons in the CN decreased. However, either the damaged neurons recovered, or axonal sprouting from the original neurons must have occurred, because the measured amplitudes were large. Moreover, with the exception of the 4 kHz frequency measurements, there was no difference in the amplitude of wave IV between the two noise exposure groups. This suggests that the levels of noise used in this study were insufficient for affecting the IC, or that neuronal activity in the CN was able to compensate for the damage.

All three rows of outer HCs in the control groups were normal, and there were no missing HCs. Only a few HCs were missing in the middle and basal turn sections in the day 1 and day 3 treatment groups. The minimum survival rates of outer HCs in the basal turn and apical turn sections were 78% and 99%, respectively. This suggests that outer HC loss was mild in general. The confirmation of outer HC loss indicates that our noise-exposure protocol can cause a PTS, while also providing evidence that as little as 2 h of noise exposure can permanently damage the cochlea [23]. When a TTS occurs, the ABR threshold returns to its normal level, but long-term damage to the synapses may exist, even in the absence of HC loss. The disruption of signaling between the inner HCs and type-1 afferent auditory nerve fibers causes degeneration of the auditory nerve fibers and spiral ganglia. In particular, if synapses connecting low-spontaneous-rate auditory nerve fibers deteriorate, communication may be disrupted due to the signals being less distinct from background noise [24]. Therefore, it is important to protect the auditory canals from noise, regardless of its intensity and duration.

Loss of HCs after noise exposure can lead to secondary damage, including auditory synaptopathy. Noise exposure not only damages the cochlea, but also triggers extensive changes in the central auditory pathway. Sensory deprivation due to a decrease in the number of HCs can induce a reduction in the cell density in the upper auditory structures. For example, after overstimulation, the cell population in the ventral CN (VCN) has been shown to decrease due to apoptosis [25].

It is important to understand the molecular events that occur after acoustic trauma, in order to minimize damage to the central auditory pathway. A bioinformatics analysis by Alagramam et al. showed that exposure to both 116 and 110 dB noise could induce genes related to the MAPK signaling pathway For example, the Fos gene, which has a putative role in neuronal apoptosis and cell death, was induced in response to both treatments [26].

Acoustic overstimulation, unlike ablation, can lead to widespread axon degeneration and death. A study of cats with acoustic trauma demonstrated that new axons are able to grow in the VCN. Following noise exposure, cochlear nerve endings in the VCN degenerated over a period of several months and disappeared completely after 3 years. However, other small axons subsequently began to appear throughout the VCN, suggesting that degenerated neurons can reorganize the structure of the CN by generating new axons [27].

The CN is the primary point of convergence between auditory and somatosensory inputs. The balance of auditory sensory inputs can be disrupted by a decrease therein due to peripheral hearing loss. This imbalance, caused by increased excitatory activity and/or decreased inhibitory activity, enhances central neural receptivity and leads to hyperexcitability [28]. This change in the plasticity of the CN occurs in the form of axonal sprouting, which is regulated by the TGF-β signaling pathway [29]. However, such axonal sprouting can trigger tinnitus, which may itself be problematic for some patients [30]. Therefore, many researchers are trying to develop treatments that prevent axonal sprouting by inhibiting the TGF-β signaling pathway.

Among the potential genes targeted by the miRNAs described here, dual specificity protein phosphatase 10 (DUSP 10; Gene ID, 11221) is co-regulated by *miR-411-3p* and *miR-183-5p*. The expression levels of both *miR-411-3p* and *miR-183-5p* were shown to decrease after acoustic trauma, leading to an increase in the DUSP 10 level. The by-products of DUSP 10 inactivate p38 and stress-activated protein kinase/c-Jun NH(2)-terminal kinase (SAPK/JNK), thereby inhibiting the JNK pathway. As the JNK pathway serves as a major driver of apoptosis, inhibition thereof is likely to protect cells against apoptosis [31].

Profilin2 (PFN2) is also co-regulated by *miR-411-3p* and *miR-183-5p*. The expression levels of both *miR-411-3p* and *miR-183-5p* were shown to decrease after acoustic trauma, leading to an increase in PFN2. PFN2 is an actin binding protein that plays an important role in maintaining the structure of synapses in neural tissues [32]. In the case of TTS, afferent synapses are damaged and an increase in PFN2 expression may promote structural recovery of the damaged synapses [33].

## 4. Materials and Methods

### 4.1. Study Design

Noise-induced hearing loss (NIHL) was achieved by exposing subjects to 2 h of noise at a 115 dB sound pressure level (SPL). Tests of the auditory brainstem response (ABR) and histological examinations of the cochleae confirmed loss of hearing. Microarray analysis of the CN and IC tissues was used to identify candidate microRNAs (miRNAs). These miRNAs were validated using quantitative reverse transcription polymerase chain reaction (qRT-PCR) and target pathway analysis.

### 4.2. Animal Subjects

All of the animal experiments described were approved (8 February 2018) by the Institutional Animal Care and Use Committee of Seoul National University Hospital (Seoul, Korea; 18-0025-C1A0), which is endorsed by the Association for the Assessment and Accreditation of Laboratory Animal Care International. The animals used in these experiments were kept under 12-h/12-h day/light cycle conditions, with free access to food and water. They were acclimated to laboratory conditions 1 week prior to the initiation of these experiments. A total of 48 male Sprague–Dawley rats, aged 6 weeks, were randomly separated into four groups (all, *n* = 12). One group was assayed 1 day after noise exposure (day 1), one group was assayed 3 days after noise exposure (day 3), and the other two groups were used as the day 1 and day 3 controls.

### 4.3. Noise-Exposure Protocol

Animals were anesthetized using a mixture of 40 mg/kg Zoletil (Zoletil 50; Virbac, Bogotá, Colombia) and 10 mg/kg xylazine (Rumpun; Bayer-Korea, Seoul, Korea) via an intramuscular injection before noise exposure. Each animal was placed in a separate wire cage to avoid unequal noise exposure, and each experiment was performed in a customized acrylic box in a sound-attenuating laboratory booth (900 mm × 900 mm× 1720 mm) with an electromagnetic shield. The animals were exposed to 2 h of broadband white noise at 115 dB SPL using a 2446-J compression driver (JBL Professional, Los Angeles, CA, USA) with an MA-620 power amplifier (Inkel, Incheon, Korea), in order to create the bilateral NIHL animal model (Appendix A). The sound intensity within the acrylic box was measured every hour using a CR152B sound level meter (Cirrus Research plc, Hunmanby, UK) to confirm that there were no alterations in the sound level during the noise-exposure treatments. The control animals were injected with the same dose of anesthetic and kept in the sound attenuating booth for the same period of time, without noise exposure [34]. Audiometry was performed at 4 h after noise-exposure treatments, to allow stable measurements to be recorded.

### 4.4. Auditory Brainstem Response (ABR) Recordings

The hearing function of all animals was evaluated before noise exposure using the ABR. Animals were anesthetized and placed in sound-attenuating booths. Subdermal needle electrodes were positioned at the nape of the neck as the vertex, the ipsilateral mastoid as the negative, and the contralateral mastoid as the ground (Appendix A) [35]. Sound stimuli tone-bursts of 4, 8, and 16 kHz (duration, 1562 µm; CoS shaping, 21 Hz) were applied. High-frequency software (ver. 3.30; Intelligent Hearing Systems, Miami, FL, USA) and high-frequency transducers (HFT9911-20-0035; Intelligent Hearing Systems) were used to measure the ABR. Before obtaining the electroencephalography signal, the impedance between the electrodes was assessed to establish whether this was less than 2 kΩ. Responses to the signal were amplified approximately 100,000-fold and band-pass filtered (100–1500 Hz). The intensity of the stimuli ranged from 90 to 20 dB SPL in 5 dB increments. A total of 512 sweeps were averaged at each intensity level. Additional ABR measurements were recorded at 4 h, and on days 1 and 3 after noise exposure. The ABR threshold was defined as the smallest stimulus intensity level that produced a visible waveform for wave II or IV.

### 4.5. Cochlear Whole-Mount Surface Preparation

Both control (*n* = 8) and noise-exposed (*n* = 8) animals were sacrificed under anesthesia. For each sample, the cochlea was detached from the temporal bone and fixed in 4% paraformaldehyde solution for 24 h at 4 °C. Fixed ears were washed three times in 1× phosphate-buffered saline (PBS) [36]. The thin layer of laminar bone covering the cochlea was trimmed under a SZ2-ILST stereomicroscope (Olympus Corporation, Tokyo, Japan) using a drill (Strong 90; Saeshin Precision, Daegu, Korea) with a 2 mm-diameter diamond burr attachment (Appendix A). A hole was created by breaking the bone between the oval and round windows using very fine forceps (Appendix A). The laminar bone was removed using a conventional 1-mm syringe needle (Appendix A). The cochlear nerve was cut and the spiral structure of the organ of Corti was isolated. Next, the stria vascularis and Reissner’s membrane were removed (Appendix A). The first turn from the top of the organ of Corti was removed using scissors. This was called the ‘apical turn’ section. A second turn was removed and called the ‘middle turn’ section and a final half-turn section was removed and called the ‘basal turn’ section (Appendix A). The sections were placed in 1× PBS solution to prevent them from drying out.

### 4.6. Outer HC Staining

Phalloidin was used to stain F-actin, and the photostable orange fluorescent Alexa Fluor 546 dye was used to visualize the cuticular plate and stereocilia within the HCs. After surface preparation, the isolated spiral structure of the organ of Corti was incubated in a mixed solution of 0.3% Triton X-100 and Alexa Fluor 546 phalloidin (1:100 dilution; Invitrogen, Carlsbad, CA, USA) for 45 min at room temperature in a lightproof box [37]. The sample was washed three times in 1× PBS and separated into three segments using Vannas capsulotomy scissors (E-3386; Karl Storz SE & Co. KG, Tuttlingen, Germany), consisting of the apical, middle, and basal turn sections. The first complete turn from the apex was the apical turn, the next complete turn was the middle turn, and the final half-turn was the basal turn. Each turn section was mounted on a slide using ProLong™ Gold Antifade mountant (P36930; Invitrogen) to prevent the fluorescent dyes from fading. Images were generated using a STED CW confocal laser scanning system (Leica, Wetzlar, Germany) and HCs within the images were counted.

### 4.7. Cochlear Histology

Cochleae from both control (*n* = 8) and noise-exposed (*n* = 8) rats were fixed and washed. Samples were decalcified using 10% (*w/v*) ethylenediaminetetraacetic acid (Santa Cruz Biotechnology, Dallas, TX, USA) in 1× PBS for 4 weeks. A histological examination was performed weekly to determine when the samples were ready for the embedding procedure. The tissues were dehydrated using a series of ethanol washes, and the ethanol was then removed using xylene. After the ethanol was removed, tissues were infiltrated with paraffin wax [38] in a PELORIS II tissue processing system (Leica). The processed cochleae were embedded in a stainless mold and trimmed into 4-µm-thick sagittal sections using a RM2255 microtome (Leica). The sections were deparaffinized for 1 h at 60 °C in a dry oven and cleaned using a series of ethanol washes. Next, the nuclei were stained for 7 min with hematoxylin (DAKO, Jena, Germany) and the cytoplasm was stained for 30 s with eosin Y (Sigma-Aldrich, St. Louis, MO, USA). The stained slides were dehydrated and preserved in 70% ethanol [39]. After mounting, the organ of Corti was examined using a light microscope (ECLIPSE Ci-L; Nikon, Tokyo, Japan).

### 4.8. RNA Extraction

Whole brain tissue was harvested, and the bilateral CN and IC were dissected out using a brain matrix. The locations of the CN (−9.30 to −11.30 mm from the bregma) and the IC (–8.30 to –9.30 mm from the bregma) were determined according to the rat brain atlas [40] (Appendix A). Tissues were frozen in liquid nitrogen immediately after removal and stored at −80 °C. The harvested tissue was lysed in 1 mL QIAzol solution using a TissueLyzerII (Qiagen, Hilden, Germany) and incubated at room temperature for 5 min. The samples were placed on a vortex mixer after adding 200 µL chloroform to each and then incubated at room temperature for 3 min. Next, the samples were centrifuged at 12,000× *g* for 15 min at 4 °C and the upper aqueous phase containing the RNA was removed to a fresh tube. A total of 500 µL isopropyl alcohol was added to each tube. The tubes were then inverted and incubated at room temperature for 10 min. Thereafter, the tubes were centrifuged at 7500× *g* for 5 min at 4 °C and the RNA pellets were washed twice with 1 mL 75% ethanol. The pellets were dried for approximately 5 min and redissolved in RNase-free water.

### 4.9. Analysis of miRNA Arrays

The analysis of miRNAs was performed by Ebiogen Inc. (Seoul, Korea) using the Affymetrix GeneChip miRNA 4.0 array (Affymetrix, Santa Clara, CA, USA). A total of 24 animals were randomly separated into two treatment groups and two corresponding control groups. Treated rats were exposed to noise and assayed after 1 or 3 days. After hearing loss was confirmed, the CN and IC from two animals from the same group were combined and treated as a single sample, and three samples from each group were used for the analysis. Extracted total RNA was assessed for quality and quantity using a Bioanalyzer 2100 system (Agilent, Santa Clara, CA, USA). A total of 250 ng of RNA was analyzed. After ligating biotin-labeled 3DNA dendrimers, each RNA strand was labeled using poly-A polymerase. The biotinylated RNA strands were hybridized for 18 h at 48 °C on an Affymetrix GeneChip miRNA 4.0 array. The hybridized GeneChip was washed and stained using an Affymetrix 450 Fluidics station. Fluorescence signals from the 3DNA dendrimers were detected using an Affymetrix GeneChip 3000 7G scanner.

### 4.10. Quantitative Reverse Transcription Polymerase Chain Reaction (qRT-PCR)

Using an miScript^®^ II RT kit (Qiagen, Hilden, Germany), 2 µg of RNA was mixed with reverse-transcription master mix and incubated for 60 min at 37 °C. To inactivate the miScript reverse transcriptase, the mixture was incubated for 5 min at 95 °C and then placed on ice. A total of 20 µL of cDNA was diluted to 1:16 and used as template cDNA. The miScript SYBR^®^ Green PCR kit (Qiagen) was used with miScript Primer Assay reagents (Qiagen) for qRT-PCR. U6 small nuclear RNA was used as an endogenous control gene [41]. The miScript Primer Assay reagents and the reaction mix were dispensed into wells containing template cDNA. The PCR plate was sealed with film and centrifuged at 1000× *g* for 1 min at room temperature. Initial activation was performed for 15 min at 95 °C. The reactions consisted of 40 cycles of denaturation, annealing, and extension, and fluorescence data were collected during the extension phase. The reactions were performed using an ABI 7500 real-time PCR system (Applied Biosystems, Foster City, CA, USA). Relative quantification values were obtained for each of the target genes using the observed cycle threshold (Ct) results and the 2^−ΔΔ*C*t^ method.

### 4.11. Pathway Analysis of Candidate miRNAs

For the CN and IC, a total of 10 and 13 candidate miRNAs were selected from the microarray analysis based on 1.5-fold changes in normalized intensity values (*p* < 0.1) respectively. Of these, five and three miRNAs, respectively, were selected following qRT-PCR validation. Using DIANA-miRPath software (ver. 3.0), a KEGG pathway analysis was performed using DIANA-microT-CDS (ver. 5.0; DIANA TOOLS, http://diana.imis.athena-innovation.gr/DianaTools/index.php?r=microT_CDS/index) with a threshold of 0.8 and a false discovery rate correction [42,43]. A total of 12 and 14 KEGG pathways were identified for the CN and IC, respectively, using a gene union module and a *p*-value threshold of 0.05 and 0.3.

### 4.12. Statistical Analyses

All data are expressed as the means ± standard error of the mean, and all data were analyzed using SPSS software (ver. 25; IBM, Armonk, NY, USA). An F-test was performed to determine whether the levels of variation within the groups were equal. After the F-test, data were analyzed using Student’s t-tests to identify significant differences between groups. A *p*-value of <0.05 was considered statistically significant.

## 5. Conclusions

Using a noise exposure animal model, we were able to show that even acute short-term noise exposure can lead to hearing loss. Changes in the ABR amplitude of wave II suggest an alteration in either synaptic transmission or the number of neuronal cells. To investigate the role of miRNAs in the central auditory pathway, CN and IC were compared in both the treatment and control groups, with microarray analysis and qRT-PCR results suggesting that *miR-200b-3p*, *miR-183-5p*, *miR-411-3p*, *miR-20b-5p*, *miR-377-3p*, *miR-92a-1-5p*, *miR-136-3p*, and *miR-26b-5p* may play key roles in the neuroplasticity of the central auditory pathway. Using the KEGG database, we found that five of these candidate miRNAs may be involved in the MAPK signaling pathway, axon guidance, and the neurotrophin signaling pathway in the CN, while an additional three candidate miRNAs may influence the MAPK signaling pathway in the IC. Further validation of these candidate miRNAs will be achieved using miRNA oligomers such as mimics and inhibitors, in order to better refine the specific signaling pathways underlying these processes. These target miRNAs, which play crucial roles in the central auditory pathway, can be used for diagnosis in the early stage of NIHL, and for treatment of the damage caused by the cellular and physiological changes after NIHL.

## Figures and Tables

**Figure 1 ijms-21-08792-f001:**
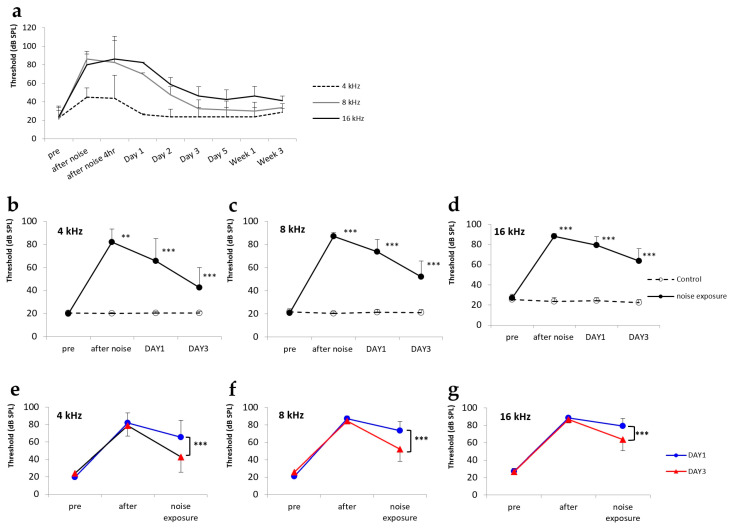
Hearing changes after noise exposure. (**a**) Long-term auditory brainstem response (ABR) data of the noise-control model obtained over a period of 3 weeks. (**b**–**d**) Line graphs showing differences between control and treatment groups at frequencies of 4, 8, and 16 kHz. *** *p* < 0.001. (**e**–**g**) Line graphs comparing ABR thresholds at days 1 and 3 after noise exposure at frequencies of 4, 8, and 16 kHz. *** *p* < 0.001.

**Figure 2 ijms-21-08792-f002:**
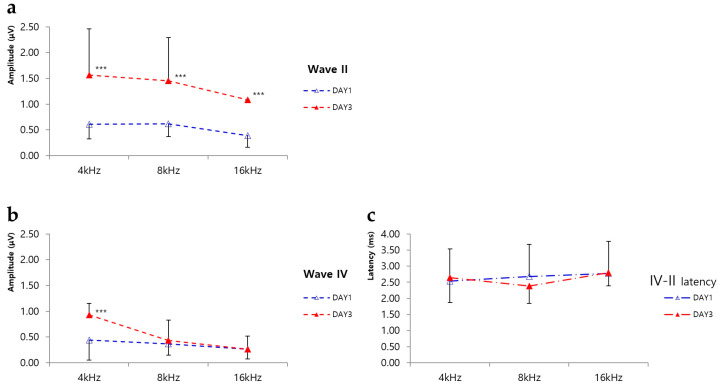
Amplitudes of waves II and IV and latencies of waves IV–II. (**a**) Comparison of the amplitude of wave II at different frequencies. The wave II amplitude for rats assayed at day 3 after noise exposure was significantly larger than that for rats assayed at day 1 after noise exposure, at all frequencies. *** *p* < 0.001. (**b**) Comparison of the amplitude of wave IV among different frequencies. The wave IV amplitude for rats assayed at day 3 after noise exposure was significantly larger than that for rats assayed at day 1 after noise exposure at 4 kHz. *** *p* < 0.001. No significant differences were observed at the other frequencies. (**c**) The latencies of waves IV–II at each frequency. No significant differences were observed at any frequency.

**Figure 3 ijms-21-08792-f003:**
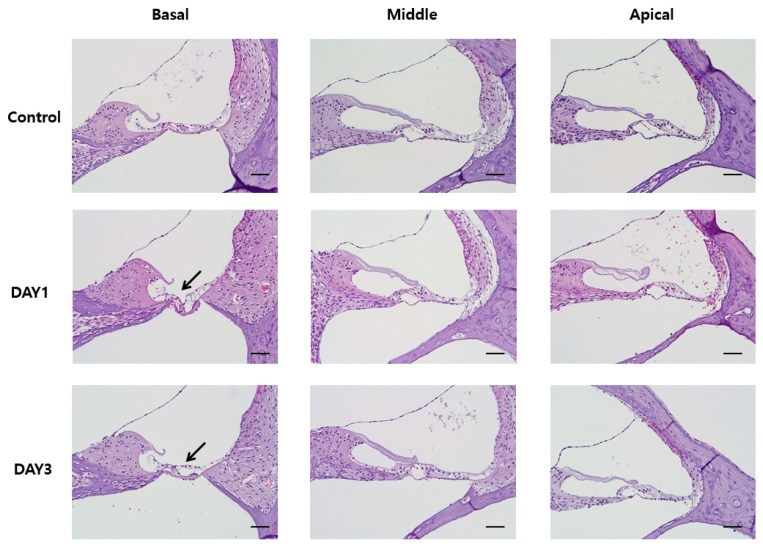
Hematoxylin and eosin (H&E) staining of the basal, middle, and apical turn sections of the organ of Corti. Damage (indicated by black arrows) was only observed in the basal turn section in rats assayed at days 1 and 3 after noise exposure. All scale bars represent 50 µm.

**Figure 4 ijms-21-08792-f004:**
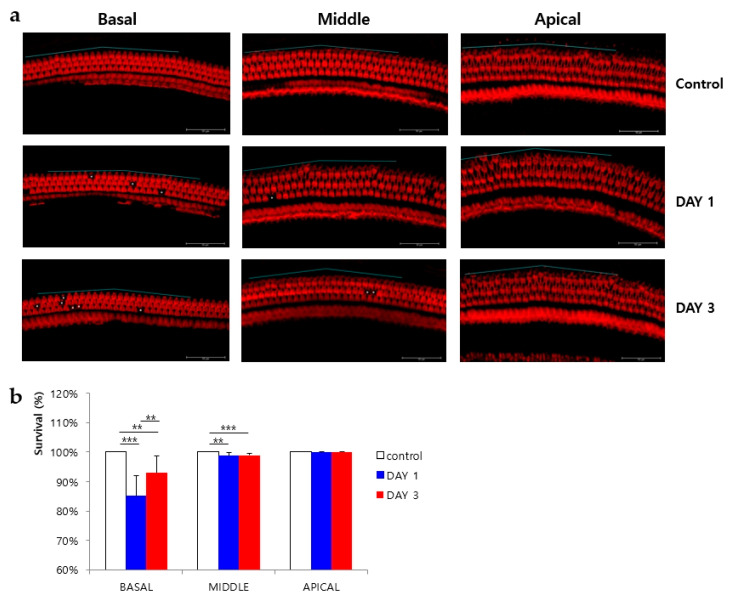
Phalloidin staining of outer hair cells (HCs) and HC survival. (**a**) Fluorescence staining of outer HCs from each turn section of the cochlea. Scale bars represent 50 µm. Asterisks indicate the positions of lost HCs. The blue line along the hair cell line indicates the length of 200 µm. (**b**) Survival rates of outer HCs in each turn section. The surviving HCs per 200 µm along the length of the cochlea in the basal, middle, and apical turn sections were counted. ** *p* < 0.05 and *** *p* < 0.001.

**Figure 5 ijms-21-08792-f005:**
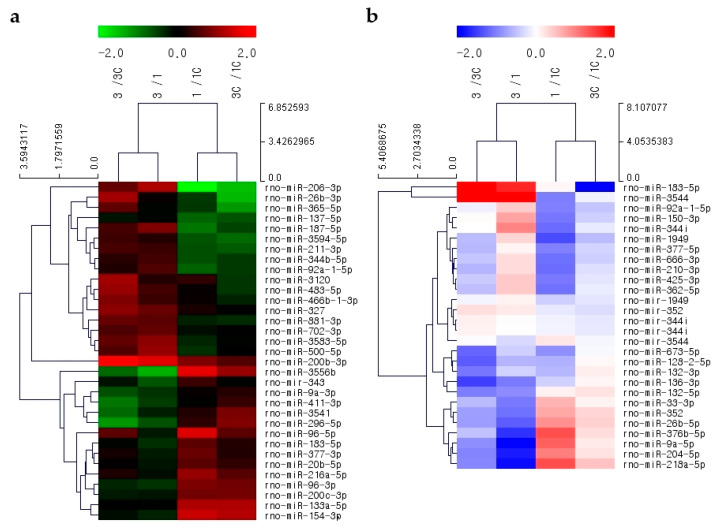
Heat maps of the CN and IC. Heat maps of (**a**) 33 miRNAs from the CN and (**b**) 27 miRNAs from the IC selected based on ≥1.5-fold changes in normalized expression (*p* < 0.1).

**Figure 6 ijms-21-08792-f006:**
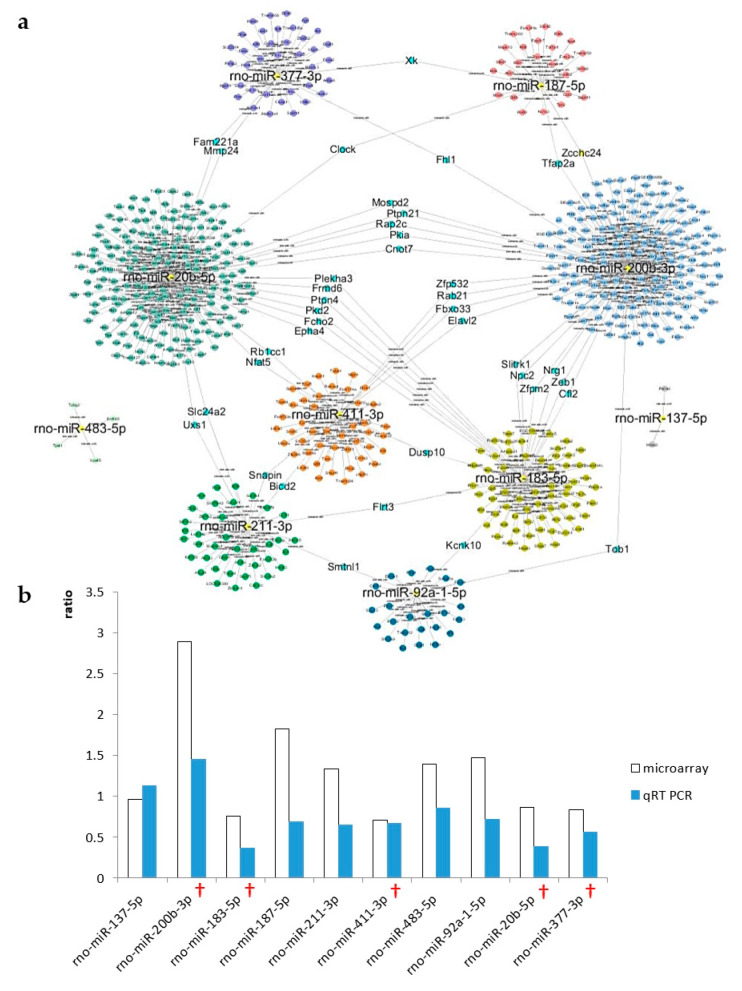
A Cytoscape map of the CN and validation of candidate miRNAs. Cytoscape was used to visualize networks among candidate miRNAs. Only miRNAs that were connected to other miRNAs were selected for validation by quantitative reverse transcription polymerase chain reaction (qRT-PCR). (**a**) A total of 10 miRNAs were selected as CN candidate miRNAs. (**b**) Ratio of expression of each candidate miRNA in the CN between the day 3 and day 1 treatment groups. Expression levels were measured using microarray analysis (open bars) and qRT-PCR (filled bars). Crosses indicate validated candidate miRNAs.

**Figure 7 ijms-21-08792-f007:**
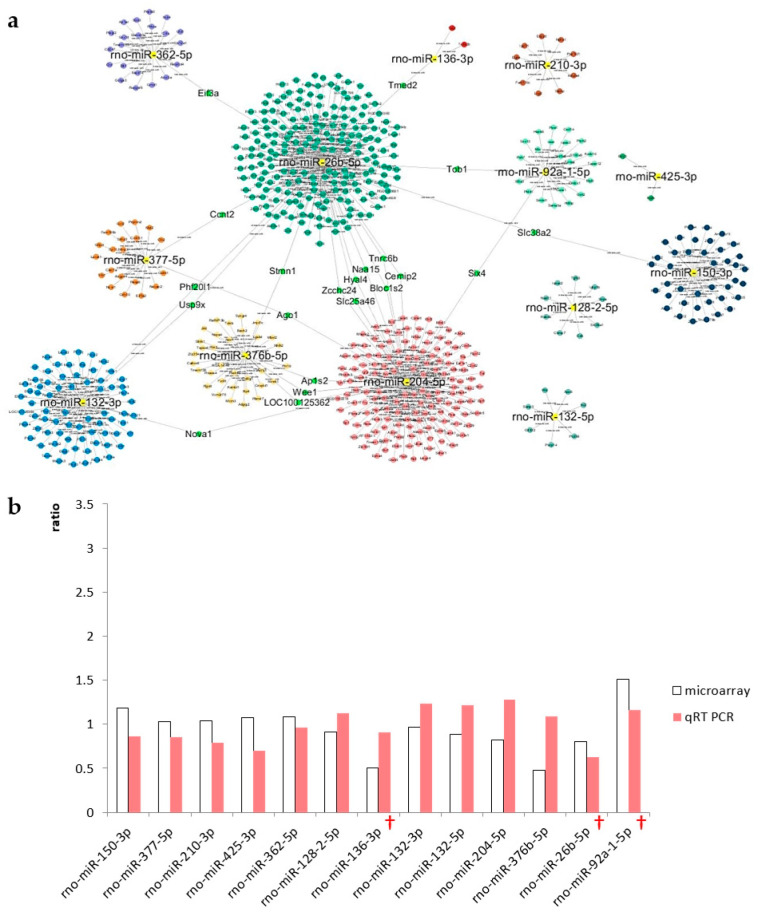
A Cytoscape map of the IC and validation of candidate miRNAs. Cytoscape was used to visualize networks among the candidate miRNAs. Only miRNAs that were connected to other miRNAs were selected for validation by qRT-PCR. (**a**) A total of 13 miRNAs were selected as IC candidate miRNAs. (**b**) Ratio of expression of each candidate miRNA in the IC between the day 3 and day 1 treatment groups. Expression levels were measured using microarray analysis (open bars) and qRT-PCR (filled bars). Crosses indicate validated candidate miRNAs.

**Figure 8 ijms-21-08792-f008:**
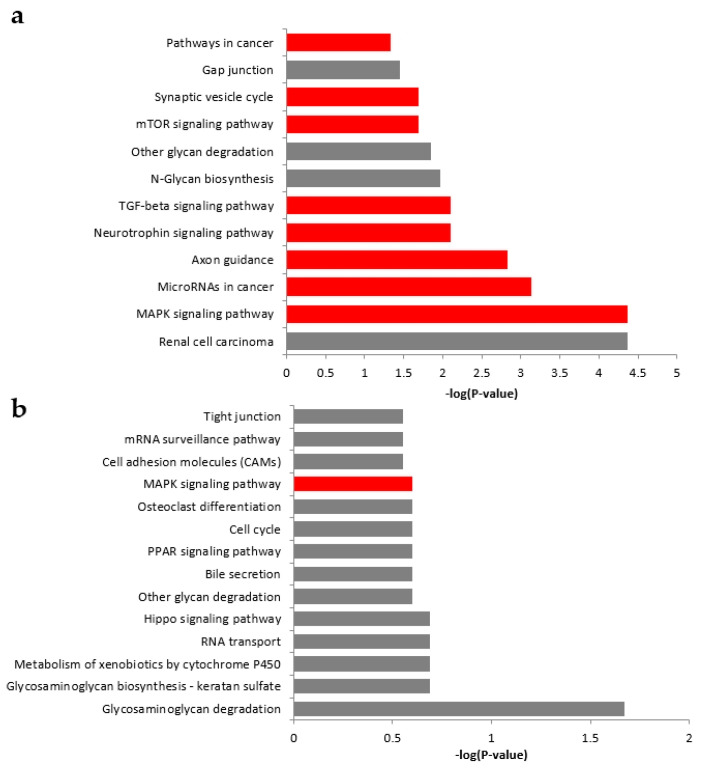
Kyoto Encyclopedia of Genes and Genomes (KEGG) pathway analysis of candidate miRNAs. (**a**) KEGG pathway analysis for the CN. Of the 12 possible in silico pathways identified, eight (marked in red) were highlighted based on their relevance to the five candidate miRNAs. (**b**) KEGG pathway analysis for the IC. Of the 14 possible in silico pathways identified, only the mitogen-activated protein kinase (MAPK) signaling pathway (marked in red) was identified as a relevant target for the three candidate miRNAs. Abbreviations: mTOR, mammalian target of rapamycin; TGF, transforming growth factor; PPAR, peroxisome proliferator-activated receptor.

**Table 1 ijms-21-08792-t001:** Candidate microRNAs (miRNAs) of the cochlear nucleus (CN).

Gene Symbol	Chromosome	Sequence Length	Sequence	1/1C ^1^	3/3C ^2^	3/1 ^3^	3C/1C ^4^
*rno-miR-411-3p*	6	20	UAUGUAACACGGUCCACUAA	0.977	0.529	0.712	1.315
*rno-miR-183-5p*	4	22	UAUGGCACUGGUAGAAUUCACU	1.542	0.957	0.759	1.222
*rno-miR-377-3p*	6	23	UGAAUCACACAAAGGCAACUUUU	1.652	1.154	0.839	1.201
*rno-miR-20b-5p*	X	23	CAAAGUGCUCAUAGUGCAGGUAG	1.522	1.029	0.870	1.288
*rno-miR-137-5p*	2	22	ACGGGUAUUCUUGGGUGGAUAA	0.576	0.871	0.968	0.640
*rno-miR-211-3p*	1	20	GGCAAGGACAGCAAAGGGGG	0.649	1.420	1.334	0.610
*rno-miR-483-5p*	1	22	AAGACGGGAGAAGAGAAGGGAG	1.072	2.025	1.393	0.737
*rno-miR-92a-1-5p*	15	23	AGGUUGGGAUUUGUCGCAAUGCU	0.588	1.194	1.468	0.724
*rno-miR-187-5p*	18	18	AGGCUACAACACAGGACC	0.529	1.404	1.827	0.688
*rno-miR-200b-3p*	5	23	UAAUACUGCCUGGUAAUGAUGAC	1.826	3.587	2.895	1.474

^1^ Fold change of day 1 treatment vs. control, ^2^ fold change of the day 3 treatment group vs. control, ^3^ fold change of the day 1 and day 3 treatment groups, and ^4^ fold change of the day 1 vs. day 3 control groups. A color index chart for of the fold change data is provided in Appendix A.

**Table 2 ijms-21-08792-t002:** Candidate miRNAs of the inferior colliculus (IC).

Gene Symbol	Chromosome	Sequence Length	Sequence	1/1C ^1^	3/3C ^2^	3/1 ^3^	3C/1C ^4^
*rno-miR-204-5p*	1	22	UUCCCUUUGUCAUCCUAUGCCU	2.020	0.511	0.299	1.181
*rno-miR-376b-5p*	6	22	GUGGAUAUUCCUUCUAUGGUUA	2.606	0.712	0.332	1.215
*rno-miR-26b-5p*	9	21	UUCAAGUAAUUCAGGAUAGGU	1.842	0.585	0.413	1.301
*rno-miR-136-3p*	6	22	CAUCAUCGUCUCAAAUGAGUCU	0.777	0.357	0.484	1.055
*rno-miR-132-5p*	10	22	ACCGUGGCUUUCGAUUGUUACU	1.085	0.491	0.534	1.180
*rno-miR-128-2-5p*	8	21	GGGGGCCGAUGCACUGUAAGA	0.650	0.412	0.658	1.039
*rno-miR-132-3p*	10	22	UAACAGUCUACAGCCAUGGUCG	0.670	0.472	0.782	1.109
*rno-miR-377-5p*	6	22	AGAGGUUGCCCUUGGUGAAUUC	0.499	0.676	1.066	0.786
*rno-miR-210-3p*	1	22	CUGUGCGUGUGACAGCGGCUGA	0.452	0.659	1.217	0.834
*rno-miR-92a-1-5p*	15	23	AGGUUGGGAUUUGUCGCAAUGCU	0.498	0.918	1.333	0.723
*rno-miR-425-3p*	8	21	AUCGGGAAUAUCGUGUCCGCC	0.479	0.749	1.352	0.865
*rno-miR-362-5p*	X	24	AAUCCUUGGAACCUAGGUGUGAAU	0.464	0.699	1.353	0.898
*rno-miR-150-3p*	1	19	CUGGUACAGGCCUGGGGGA	0.474	1.023	1.669	0.774

^1^ Fold change of day 1 treatment group vs. control, ^2^ fold change of the day 3 treatment group vs. control, ^3^ fold change of the day 1 vs. day 3 treatment groups, and ^4^ fold change of the day 1 vs. day 3 control groups. A color index chart for the fold change data is provided in Appendix A.

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
