# Peer review of "Changes in microRNA Expression in the Cochlear Nucleus and Inferior Colliculus after Acute Noise-Induced Hearing Loss"

_ijms, 2020, doi:10.3390/ijms21228792_

Round 1
Reviewer 1 Report
In general it is very interesting manuscript . choice of Journal also is in my opinion appropriate. In introduction there is need for improvement and more explanation from audiological point of view - PTS and TTS terms should be applied in translation for humans. Results are prepared very nice. Scientific loudness is good. In conclusion there is need to more precise show impacts for clinical practice. How it influence to some human studies with NIHL
Author Response
Point 1: In introduction there is need for improvement and more explanation from audiological point of view - PTS and TTS terms should be applied in translation for humans.
Response 1: In the second and third paragraph of the introduction, I added a part that explains the definition and the meaning of TTS and PTS and how it affects the auditory processing and what kind of other diseases they cause.
Point 2: In conclusion there is need to more precise show impacts for clinical practice. How it influence to some human studies with NIHL.
Response 2:By adding an example of clinical trials of miRNAs at the end of the fifth paragraph of the introduction, I also rewrote the conclusion explaining about how miRNAs can be used in the clinical field. For example, a miRNAs detection kit can be used as an early diagnostic tool and also miRNAs can be used as gene therapeutics for NIHL patients.

Reviewer 2 Report
REVIEW
In an article Sohyeon Park, Seung Hee Han, Byeong-Gon Kim, Myung-Whan Suh, Jun Ho Lee, Seung Ha Oh and Moo Kyun Park “MicroRNA Expression Changes in Cochlear Nucleus and Inferior Colliculus after Acute Noise-induced Hearing Loss”.
The article by Sohyeon Park and co-authors is investigated the role of microRNAs (miRNAs) in 21 the neural plasticity of the central auditory pathway after acute Acute Noise-induced Hearing Loss.
Major comment
No major comments, this is a perfect scientific work.
Minor comments
List of references not correctly framed, numbers are duplicated.
Author Response
Point 1:List of references not correctly framed, numbers are duplicated.
Response 1: Since I rewrote some parts of the introduction, some of the references were removed and some references were added. Therefore, the reference numbers were rearranged. I double checked for the duplicated references and edited.
